# AirSketch: Generative Motion to Sketch

**Hui Xian Grace Lim** [*]
hxgrace@ucf.edu

**Xuanming Cui** [*]
xu979022@ucf.edu

**Yogesh S Rawat**
yogesh@crcv.ucf.edu

**Ser-Nam Lim**
sernam@ucf.edu

University of Central Florida

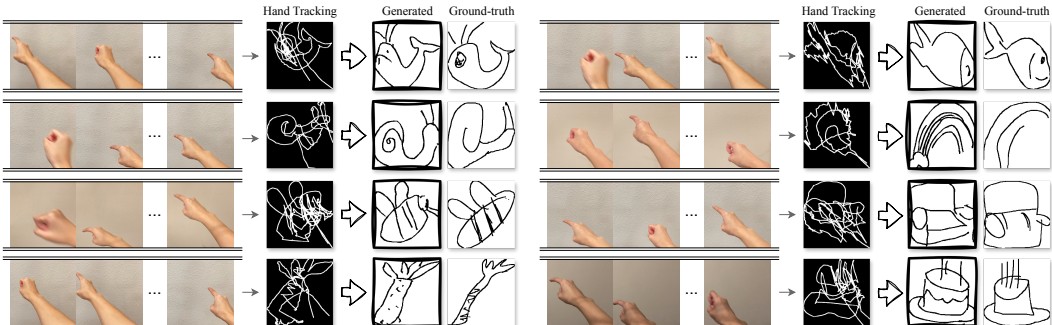

Figure 1: Given a hand drawing video, we first use a hand detection algorithm to extract the tracking image (Hand Tracking). We then take this noisy tracking image and generate a clean sketch (Generated), which faithfully and aesthetically resembles the intended sketch (Ground-truth).

## Abstract

Illustration is a fundamental mode of human expression and communication. Certain types of motion that accompany speech can provide this illustrative mode of communication. While Augmented and Virtual Reality technologies (AR/VR) have introduced tools for producing drawings with hand motions (air drawing), they typically require costly hardware and additional digital markers, thereby limiting their accessibility and portability. Furthermore, air drawing demands considerable skill to achieve aesthetic results. To address these challenges, we introduce the concept of AirSketch, aimed at generating faithful and visually coherent sketches directly from hand motions, eliminating the need for complicated headsets or markers. We devise a simple augmentation-based self-supervised training procedure, enabling a controllable image diffusion model to learn to translate from highly noisy hand tracking images to clean, aesthetically pleasing sketches, while preserving the essential visual cues from the original tracking data. We present two air drawing datasets to study this problem. Our findings demonstrate that beyond producing photo-realistic images from precise spatial inputs, controllable image diffusion can effectively produce a refined, clear sketch from a noisy input. Our work serves as an initial step towards marker-less air drawing and reveals distinct applications of controllable diffusion models to AirSketch and AR/VR in general. Code and dataset are available under: `https://github.com/hxgr4ce/DoodleFusion`.

---

[*] Authors contributed equally.

38th Conference on Neural Information Processing Systems (NeurIPS 2024).

# 1  Introduction

Hand gestures are an essential element in communication[43]. In particular, iconic hand motions (*i.e.* air drawing) can depict visual aspects of an object. This form of expression is frequently used to visually supplement verbal communication and is used in various practical applications, including conceptual discussions, overcoming language barriers, and aiding in visual design.

Popular AR/VR tools like Google's Tiltbrush [20] and HoloARt [5] create visuals via hand motions, but are inconvenient. These applications generally require head-mounted displays and digital hand markers, which are costly and hinder portability. Furthermore, their weight and temperature make them unsuitable for continuous use [60, 72, 13], and so are inconvenient for spontaneous usage. Yet, these devices provide accurate positioning, stabilization, and varied brush controls and are crucial to producing high-quality drawings.

*Can we generate sketches from hand motions without additional sensors or markers?* In order to enhance accessibility and convenience, we aim to generate sketches from hand motions videos captured using any standard camera embedded in devices like smartphones or smart glasses.

One could clearly deploy hand tracking algorithms [9, 74] to turn these hand motion videos into sketches. However, creating air drawings with a hand tracking algorithm alone presents several challenges. These include the user's drawing ability, physical fatigue, and inaccuracies in hand tracking. Noise in hand tracking can severely distort a sketch, rendering it almost unrecognizable.

The objective, therefore, is to generate clean sketches that faithfully represent the user's intent, from highly noisy and distorted hand motion input. This task requires the model to possess a deep understanding of shape and object priors, enabling it to discern and correct deformed motion cues while filtering out undesirable noise. We refer to this task as Generative Motion to Sketch.

There are many approaches to this task, with different architecture and data modalities. The input modality might include learned video representations, coordinate sequences from a hand tracking algorithm, or a rasterized image. Depending on the modality, the task may also be reformulated as video-to-sketch, sequence-to-sequence, image-to-image, or a combination thereof. This diversity introduces interesting opportunities for rich exploration of all these diverse approaches.

We explore the use of controllable image Diffusion Models (DM) in generating sketches from motion. Existing work such as ControlNet [76] and T2IAdapter [44] generate photo-realistic images given spatially-precise conditioning images. We explore a different use case by using controllable DMs to "reconstruct" clean sketches from severely distorted and noisy input images obtained with a hand tracking algorithm. We propose a simple, augmentation-based, self-supervised training procedure and construct two air drawings datasets for evaluation purposes.

Our experiments show that with our augmentation-based training, controllable image DMs are able to recognize and interpret correct visual cues from noisy tracking images, some of which even appear to be nearly unrecognizable to the human eye, and generate sketches faithfully and aesthetically, while being robust to unseen objects. Moreover, we show that through simple augmentations, the model is capable of sketch-completion and text-instructed stroke styling. Finally, we conduct ablations to investigate 1) the effects of different augmentations, 2) the contribution of text prompts to sketch generation and 3) the effect of different levels of input 'chaos' on the quality of resulting generations.

In summary, in this paper:

1. We conduct a pilot study of AirSketch, sketch generation from marker-less air drawing, and provide two air drawing datasets for evaluation.

2. We propose a controllable DM approach that generates faithful and aesthetic sketches from air-drawn tracking images with a self-supervised, augmentation-based training procedure, and conduct ablation studies to prove its effectiveness and robustness.

3. We explore a different way of using spatially-conditioned DMs and reveal some of their interesting properties in the context of sketch generation. We hope our experiments shed new light on understanding the properties of controllable DMs and facilitate future research.

## 2 Related Works

### 2.1 Sketching in AR/VR

There are many methods for drawing in AR and VR. Applications such as Google's Tilt Brush [20], Mozilla's A-Painter [45], Oculus' Quill [48], and HoloARt [5] display user strokes as lines or tubes that can extend in all three dimensions. Many sketching applications such as these require a combination of VR/AR headsets and controllers to use. Since drawing freehand with six degrees of freedom makes it difficult to draw steady lines and surfaces, applications like AdaptiBrush [59] and Drawing on Air [28] use trajectory-based motion of ribbons to render strokes predictably and reliably. Just-a-Line [21] and ARCore Drawing [26] are smartphone-based AR drawing applications, where users draw lines by moving a smartphone in the air [21], or drawing on a smartphone screen [26].

### 2.2 Sketch Applications

**Representing Sketches.** A sketch can be represented in both vector and pixel space, and with varying levels of abstraction. A sketch can be represented as a rasterized image [40, 75, 50, 37, 32], a sequence of coordinates [23, 56, 38], a set of Bezier curves [67, 18], or a combination [8]. These representation methods are suited for different tasks. For example, the rasterized image representation is commonly used in Sketch-Based Image Retrieval (SBIR) in order to compare sketches with images, while the coordinate sequence representation is often used for sketch generation. On the other hand, a sketch can also depict the same object at varying abstraction levels, thereby imposing further challenges on downstream tasks, especially sketch-based retrieval in 2D [37, 32, 62] and 3D [69, 41, 14, 31].

**Sketch Generation.** Most existing sketch generation methods adopt the vector representation [23, 56, 38], and view sketch generation as an auto-regressive problem on coordinate sequences. These sequences are typically represented by lists of tuple $(\delta x, \delta y, \mathbf{p})$, where $\delta x$ and $\delta y$ represent the offset distances of x and y from the previous point, and $\mathbf{p}$ is a one-hot vector indicating the state of the pen. Sketch-RNN [23] uses a Variational Autoencoder (VAE) with a bi-directional RNN as the encoder and an auto-regressive RNN as the decoder. Sketch-Bert [38] follows the BERT [15] model design and training regime. Instead of auto-regressive sketch generation, SketchGAN [39] takes in a rasterized sketch image and uses a cascade GAN to iteratively complete the sketch. Sketch-Pix2Seq [12] adds a vision encoder on top of Sketch-RNN and thus allow the model to take in an image input and reconstruct the sketch using coordinate sequences.

Nonetheless, these methods generate sketches in the input sketch modality by taking in a sketch and reconstructing the exact same sketch, or by predicting endings given incomplete sketches. In contrast, our work considers the task for generating sketches from hand motions. Specifically, we adopt the image representation for sketch generation as opposed to using coordinate sequences. This offers several advantages: it 1) maintains constant computation complexity with regards to sketch length, 2) is drawing-order invariant, which is especially favorable in our case as we consider extremely noisy sketches as input, and 3) allows us to utilize large pretrained image generative models.

### 2.3 Image Diffusion Models

**Diffusion Probabilistic Model.** First introduced by Sohl-Dickstein *et al.* [64], diffusion probabilistic models have been widely applied in image generation [24, 65, 29]. To reduce computational cost, Latent Diffusion Models [57] project input images to lower dimension latent space, where diffusion steps are performed. Text-to-image diffusion models [47, 55, 54, 61, 51] achieve text control over image generation by fusing text embeddings obtained by pre-trained language models such as CLIP [52] and T5 [53] into diffusion UNet [58] via cross attention.

**Controllable Image Generation.** Beyond text, various recent works have focused on allowing more fine-grained controls over the diffusion process. This can be done via prompt engineering [73, 36, 78], manipulating cross-attention [6, 10, 27, 71], or training additional conditioning adapters [44, 76]. ControlNet [76] learns a trainable copy of the frozen UNet encoder and connects it to the original UNet via zero convolution. T2IAdapter [44] trains a lightweight adapter to predict residuals that are added to the UNet encoder at multiple scales.

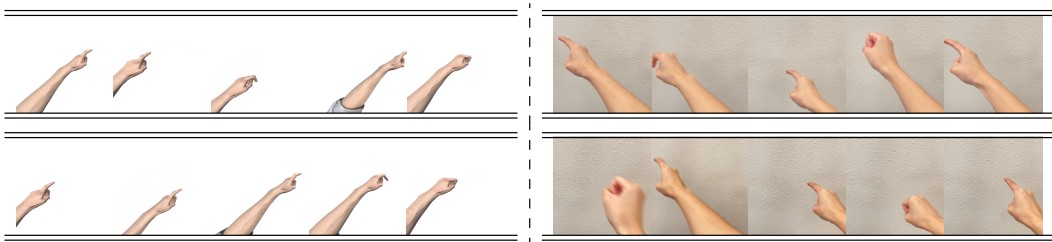

Figure 2: Examples of air drawing videos. Left: synthetic hand drawing; Right: real hand drawing.

In particular, both ControlNet and T2IAdapter take spatially-precise conditional images such as canny edge, depth, and skeleton images extracted from the original image. Koley *et al.* [33] considers human sketches as a condition and observes that human sketches have shapes that are deformed compared to canny edge or depth images, resulting in generations that lack photo-realism. Therefore, they avoid using the spatial-conditioning approach by training a sketch adapter to transform the sketch into language tokens that replace the text embeddings as the input to the cross attention.

In our work, we show that the spatial-conditioning does not have to be a "hard" constraint, at least not in the domain of sketch generation. Through proper augmentations, we discover that we can indeed teach ControlNet to map from a noisy input condition to a clean output.

### 2.4 Hand tracking and gesture recognition

Hand tracking and gesture recognition are used for several purposes, including communication and interaction in an AR environment [42] [34] and VR [1], for image-based pose estimation in sign language recognition [7] [3] [4], and many others [35] [11]. Many of these require depth-sensing hardware, such as work done by Oikonomidis et. al. using a Kinect sensor [49], or use deep learning for pose estimation [68] [19], making it difficult to integrate them into lightweight systems.

Hand pose estimation such as MediaPipe Hands [74] and OpenPose [9] take in RGB images or video and return the coordinates of 21 landmarks for each hand detected, and MediaPipe Hands is lightweight enough to integrate into even mobile devices.

## 3 Air-Drawing Dataset

In order to thoroughly evaluate our model, we need datasets with sketch-video pairs. Popular sketch datasets include Sketchy [63], TUBerlin [17], and Quick, Draw! [22]. There are also hand motion datasets that associate hand motions with semantic meaning, such as Sign Language MNIST [66], How2Sign [16], and the American Sign Language Lexicon Video Dataset [46]. However, there are no datasets that associate sketches with air drawing hand motions, prompting us to create our own sketch-video pair datasets for evaluation purposes.

**Synthetic Air-Drawing Dataset.** We use samples from the Quick, Draw! dataset [22] as the intended ground-truth sketch; each stroke is represented as a sequence of timestamps and coordinates. A 3D human arm asset is animated in the Unity engine [30] using inverse kinematics and rotation constraints, see Figure 2 (left). While following stroke coordinates, the index finger is extended, and when the stroke ends, the hand is in a closed fist. The videos have an aspect ratio of 1920 by 1080 pixels and were recorded at 60 frames per second. We choose 50 common object categories from Quick, Draw! dataset, each with 100 sketches to form our synthetic dataset with a total of 5000 sketch-video pairs. This synthetic Air-Drawing dataset simulates the scenario where users have perfect drawing ability, and the errors are solely introduced by the tracking algorithm.

**Real Air-Drawing Dataset.** A human user attempts to replicate sketches from the Quick, Draw! dataset by moving their index finger through the air. The videos were recorded with aspect ratio of 1280 by 720 pixels and at 30 frames per second. We take 10 samples per category used in the synthetic dataset, resulting in 500 video-sketch pairs.

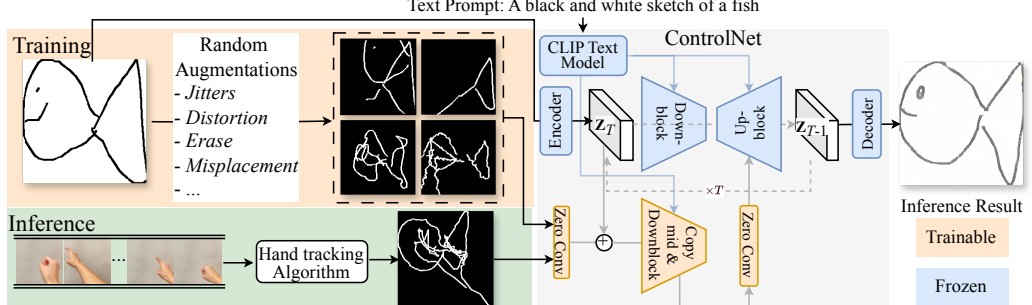

Figure 3: The overall pipeline for training and inference. During training, we randomly apply augmentations to the original ground-truth sketch to form a distorted view. The distorted view is passed to ControlNet, which is asked to generate the original, undistorted sketch. During inference, a hand-tracking algorithm is used on a hand motion video to create the input.

## 4 Methods

Our method trains a controllable DM to recover the clean sketch from a noisy one, which we discuss in Sections 4.1 and 4.2. As input at training time, we simulate noisy sketches produced from the direct application of a hand tracking algorithm, discussed in Section 4.3. Finally, we evaluate our model using two different datasets: a dataset of 3D animated hand motion videos, and a small dataset of real hand motion videos. The creation of these two datasets is discussed in Section 3.

### 4.1 Preliminary: Controllable DMs

Diffusion models involve a forward and inverse diffusion process. Given an input image $x_0$, the forward process gradually adds noise to $x_0$ to form a noisy input $x_t$ at time step $t$ as:

$$x_t = \sqrt{\bar{\alpha}_t}x_0 + (\sqrt{1 - \bar{\alpha}_t})\epsilon, \epsilon \sim \mathcal{N}(0, I) \tag{1}$$

where $\bar{\alpha}_t := \prod_{i=1}^{t} \alpha_i$, and $\alpha_t = 1 - \beta_t$ is determined by a pre-defined noise scheduler [24].

The reverse process trains a denoising UNet $\epsilon_\theta(\cdot)$ to predict the noise $\epsilon$ added to input $z_0$. In the context of controllable generation, as in ControlNet [76] and T2IAdapter [44], with a set of conditions including a text prompt $c_t$ and an additional condition $c_f$, the overall loss can be defined as:

$$\mathcal{L} = \mathbb{E}_{x_0, t, c_t, c_f, \epsilon \sim \mathcal{N}(0,1)} \left[ \|\epsilon - \epsilon_\theta(x_t, t, c_t, c_f)\|_2^2 \right], \tag{2}$$

### 4.2 Training Controllable DMs for Sketch Recovery

We adopt ControlNet [76] as our primary controlling approach. Our training procedure is illustrated in Figure 3. Due to the lack of sketch-video pair datasets, we devise a self-supervised, augmentation-based training procedure. During training, for each sketch image, we randomly sample combinations of augmentations $\mathcal{A}(\cdot)$ and apply to $x_0$ to get the distorted view $\mathcal{A}(x_0)$. It is then used as the input to ControlNet's conditioning adapter. Hence, the loss function 2 can be re-written as:

$$\mathcal{L} = \mathbb{E}_{x_0, t, c_t, c_f, \epsilon \sim \mathcal{N}(0,1)} \left[ \|\epsilon - \epsilon_\theta(x_t, t, c_t, \mathcal{A}(x_0))\|_2^2 \right]. \tag{3}$$

Therefore, unlike regular controllable DMs where the conditioning adapter takes in edge-like maps and predicts spatial-conditioning signals to be injected to the UNet, our adapter learns both the spatial-conditioning signals and a mapping from the distorted to the clean input: $\mathcal{A}(x_0) \to x_0$.

### 4.3 Sketch Augmentations

We categorize the prevalent errors from air drawings into three types: 1) user-induced artifacts such as hand jitters and stroke distortions, 2) hand tracking errors such as inaccurate hand landmark predictions, unintended strokes, and 3) aesthetic shortcomings related to the user's drawing proficiency.

In order to closely replicate these artifacts, we carefully examine typical noise found in real tracking samples and divide them into 3 categories: local, structural, and false strokes. For each category, we observe several types of artifacts, and apply augmentations to introduce each artifact to an input sketch. Visual examples for these augmentations are shown in Figure 4.

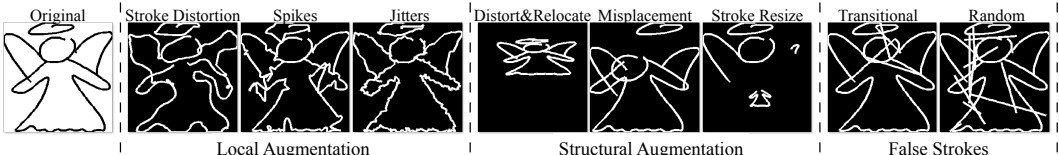

Figure 4: Visual examples of different augmentations.

**Local artifacts** include jitters, stroke-wise distortion, and random spikes. Jitters are common and become especially obvious when drawing speed is low. Stroke-wise distortion happens when users fail to properly draw the desired shape primitive (e.g. drawing a rectangle but ending up with unwanted curvy edges), potentially due to lack of friction and visual landmarks. Random spikes can arise from an accidental jerk of a user's hand or an incorrect detection from the tracking algorithm.

**Structural artifacts** include sketch-level distortion, incorrect stroke size, and misplacement. Sketch-level distortion refers to an overall distorted aspect ratio; incorrect stroke size and misplacement occur when the user unintentionally draws a stroke too small/large, and/or at the wrong position.

**False strokes** refer to unintentionally drawn strokes which commonly occur in three situations: entering or exiting the canvas, transitioning between strokes, and hesitating during drawing.

Unlike real images where shapes and scales are strictly defined, sketches are more versatile and thus do not have a clear boundary for correctness. For example, a slight change of scale during augmentation does not falsify the sketch. Therefore, we carefully tune the augmentation parameters such that the resulting augmented sketches are, in general, obviously incorrect.

# 5 Experiments

**Datasets and Implementation Details.** In training, we use a subset of 100 categories from the Quick, Draw! dataset [22]. Because a large portion of Quick, Draw! sketches are not well-drawn, we first calculate the CLIP Image-Text similarity between each sketch and their corresponding category and select the top 5% from each category, resulting in 60K sketches. Note that the sketches used for training are mutually exclusive with the sketches used for generating synthetic and real tracking images used during evaluation. To test for generalizability, we select 10 categories with similar statistics [2] from the rest and exclude them from training.

We primarily use Stable Diffusion XL [51] (SDXL) in our experiments, and adhere to the original ControlNet training and inference procedures. During both training and inference phases, we use text prompts in the format of "a black and white sketch of a <category>" to guide the model generation. We also finetune SDXL on the Quick, Draw! dataset with Low-Rank Adaptation [25] (LoRA) in order to "equip" the model with the basic ability to generate sketches in the appropriate style.

**Evaluation Metrics.** We primarily focus on using faithfulness, or the similarity between the generated sketch and the ground-truth sketch, as our model performance. Due to the versatility of sketches, we adopt multiple metrics to ensure comprehensive measurements. On the pixel-level, we use SSIM [70] to measure detailed local structural similarity, and Chamfer Distance [2] (CD) for global comparison, as CD is less sensitive to local density mismatch. Taking a perceptual perspective, we adopt LPIPS [77], CLIP [52] Image-to-Image similarity (I2I), and CLIP Image-to-Text similarity (I2T) between sketches and their corresponding text prompts to measure "recognizability".

We benchmark our model on the similarity between the ground-truth sketch and the hand tracking image. We then train a ControlNet on sketches but without any augmentation as our second baseline.

---

[2]Details in Appendix A.2

Table 1: Results on the similarity between generated and ground-truth sketches from Quick, Draw! dataset. "Tracking" refers to hand tracking images, and "Gen." refers to generated images. "w/ Aug." refers to whether sketch augmentations have been applied. "CLIP I2I/I2T" refers to CLIP Image-to-Image/Image-to-Text similarity.

| | Dataset | Backbone | w/ Aug. | SSIM (↑) | CD (↓) | LPIPS (↓) | CLIP I2I (↑) | CLIP I2T (↑) |
|---|---|---|---|---|---|---|---|---|
| | | | | Seen Categories | | | | |
| Tracking | synth. | – | – | 0.59 | 20.12 | 0.36 | 0.80 | 0.22 |
| Gen. | synth. | SDXL | ✗ | 0.59 | 20.11 | 0.37 | 0.79 | 0.23 |
| Gen. | synth. | SD1.5 | ✓ | 0.60 | 17.98 | 0.35 | 0.80 | 0.26 |
| Gen. | synth. | SDXL | ✓ | **0.64** | **17.39** | **0.33** | **0.85** | **0.28** |
| Tracking | real | – | – | 0.55 | 32.36 | 0.42 | 0.76 | 0.21 |
| Gen. | real | SDXL | ✗ | 0.55 | 31.99 | 0.41 | 0.79 | 0.21 |
| Gen. | real | SD1.5 | ✓ | 0.59 | 27.59 | 0.38 | 0.80 | 0.27 |
| Gen. | real | SDXL | ✓ | **0.64** | **25.46** | **0.36** | **0.84** | **0.29** |
| | | | | Unseen Categories | | | | |
| Tracking | synth. | – | – | 0.59 | 20.47 | 0.36 | 0.80 | 0.22 |
| Gen. | synth. | SDXL | ✗ | 0.59 | 20.32 | 0.35 | 0.81 | 0.22 |
| Gen. | synth. | SD1.5 | ✓ | 0.60 | 17.50 | 0.35 | 0.80 | 0.26 |
| Gen. | synth. | SDXL | ✓ | **0.64** | **17.27** | **0.34** | **0.85** | **0.27** |
| Tracking | real | – | – | 0.54 | 33.92 | 0.42 | 0.76 | 0.21 |
| Gen. | real | SDXL | ✗ | 0.55 | 33.53 | 0.41 | 0.78 | 0.21 |
| Gen. | real | SD1.5 | ✓ | 0.61 | 27.67 | 0.38 | 0.80 | 0.27 |
| Gen. | real | SDXL | ✓ | **0.63** | **24.26** | **0.38** | **0.85** | **0.28** |

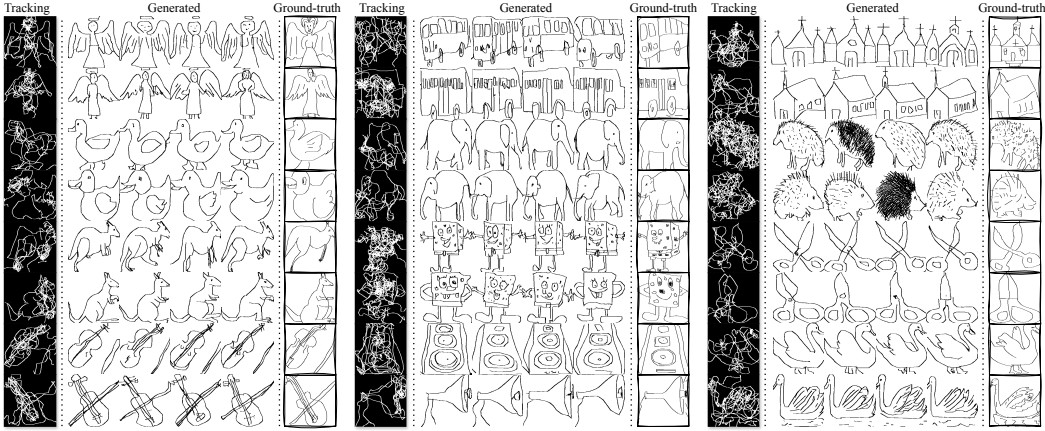

Figure 5: Generations on TUBerlin dataset.

## 5.1 Results and Analysis

**Faithfulness.** In Figure 1 we can clearly observe the ControlNet trained with augmentations successfully identifies the visual cues from the noisy tracking image, removes artifacts, and generates the clean sketches, while being aesthetic and semantically coherent. Unsurprisingly, ControlNet trained without augmentations fails to make any improvement from the tracking. In Figure 5 we show additional results where the model is trained on TUBerlin [17] dataset.

Table 1 shows quantitative results for measuring the faithfulness of generated sketch images to ground-truth. For both synthetic and real datasets, we observe a noticeable performance gain. For example, with the real dataset and SDXL, SSIM increases by 10% in SSIM, LPIPS decreases by 6%, and CD decreases by 21%.

In Table 2 and Figure 6 we show comparison between ours and Sketch-Pix2Seq [12] (P2S). Since P2S performs best when the model is trained on only one category, we randomly pick 10 categories

Table 2: Quantitative comparison between Sketch-Pix2Seq (P2S.) and ours on a subset of 10 classes.

| | SSIM (↑) | CD (↓) | I2I (↑) | I2T (↑) |
|---|---|---|---|---|
| Tracking | 0.5 | 32.36 | 0.76 | 0.21 |
| P2S. | 0.58 | 30.19 | 0.82 | 0.26 |
| Ours | **0.63** | **25.45** | **0.83** | **0.29** |

Figure 6: A comparison between generations from Sketch-Pix2Seq (top) and ours (bottom).

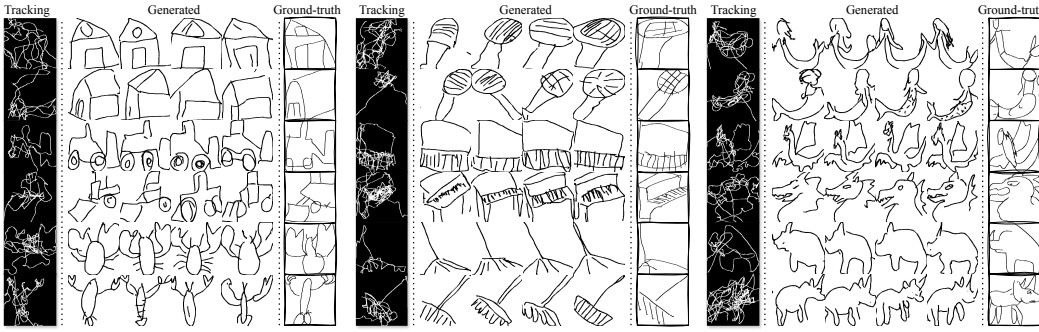

Figure 8: Generations on unseen categories from Quick, Draw! dataset.

to train 10 P2S models, and compare with ours using the selected categories. For each P2S model, following the original work, we first train 60M steps for reconstructing the exact same input sketch, and subsequently train 40M steps for constructing the clean sketch given the noisy tracking image. In Table 2 our SSIM and CD are noticeably higher than P2S, suggesting better faithfulness. This is validated in Figure 6, where we can observe while P2S is able to generate semantically correct sketches, it fails to follow the input tracking faithfully. Moreover, as P2S requires to train a separate model on each individual category and has no generalizability, it is hard to adapt to real-world usage.

In Figure 7 we show visualizations of the ControlNet hidden states across denoising steps for both the baseline model, trained with the original ControlNet recipe (top row), and our model with noise-augmented training (bottom row). The baseline ControlNet hidden states closely approximate the input edge structure early in the denoising process but fail to converge toward a clean sketch representation, resulting in largely static visualizations.

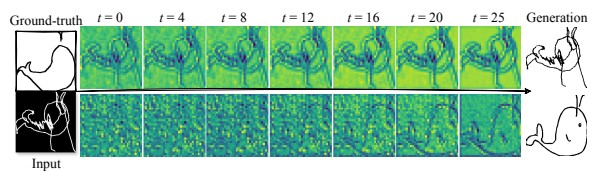

Figure 7: A comparison of visualization of ControlNet hidden states throughout denoising process from baseline approach without augmentation (top) and with augmentations (bottom).

In contrast, the hidden states from our noise-augmented training progressively reveal a coherent and accurate sketch outline throughout the denoising process.

**Generalizability.** From Figure 8 we can see that our model generalizes well to categories which are not seen during training, suggesting that the trained ControlNet learns a robust category-agnostic mapping from the noisy to clean sketch. The generalizability is also verified in Table 1, where the resulting scores for unseen categories are close to scores obtained on the seen categories.

## 5.2 Ablations

**Role of Augmentations.** In Table 3 we provide similarity scores between ground-truth and generated sketches when different combinations of augmentations are applied in training. We observe that local augmentations plays an important role in detailed local similarity: with only local augmentation applied, SSIM increases more than other metrics, with a 10% improvement to the baseline, yet increases by only 4% and 0% when only false strokes and structural augmentations are applied. Conversely, false strokes and structural augmentation have larger effects on the global similarity of the

Table 3: Model trained on seen categories with different combinations of augmentations. All the experiments are conducted using Stable Diffusion1.5 and trained for 5K steps.

| Local | Structural | False Strokes | SSIM (↑) | CD (↓) | LPIPS (↓) | CLIP I2I (↑) | CLIP I2T (↑) |
|:-:|:-:|:-:|:-:|:-:|:-:|:-:|:-:|
| | Orig. vs Tracking | | 0.55 | 32.36 | 0.42 | 0.76 | 0.21 |
| ✓ | | | 0.60 | 31.49 | 0.39 | 0.83 | 0.27 |
| | ✓ | | 0.55 | 31.14 | 0.41 | 0.80 | 0.24 |
| | | ✓ | 0.57 | 29.04 | 0.40 | 0.82 | 0.24 |
| ✓ | ✓ | | 0.60 | 31.21 | 0.39 | 0.83 | 0.28 |
| ✓ | | ✓ | 0.61 | 29.82 | 0.37 | 0.83 | 0.28 |
| | ✓ | ✓ | 0.60 | 28.97 | 0.39 | 0.82 | 0.23 |
| ✓ | ✓ | ✓ | **0.62** | **27.33** | 0.37 | **0.84** | **0.29** |

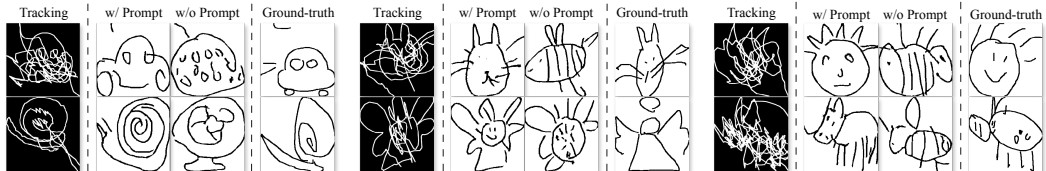

Figure 10: Examples of incorrect generation on unseen categories due to the absence of text prompt (w/o Prompt), comparing to correct generations when prompt is present (w/ Prompt).

generated sketch: CD decreases by 9% and 3% when only false strokes and structural augmentations are applied, respectively.

In Figure 9 we provide visual results when different combinations of sketch augmentations are applied during training. We observe that local augmentations are indeed crucial to removing jitters and correcting deformed lines, while false stroke augmentations ensure that the model does not falsely follow the spatial-conditions introduced by these false strokes. The structural augmentations are less significant, likely because structural artifacts are not as common as local artifacts and false strokes.

**Effect of Text Prompts.** Input tracking image conditions are extremely noisy and may not even possess obvious visual cues about the nature of the intended sketch. It is then important to investigate the effect of text guidance on the generated output and examine if our augmentation-based training significantly contributes to the generation, or if it is guided purely by text prompts.

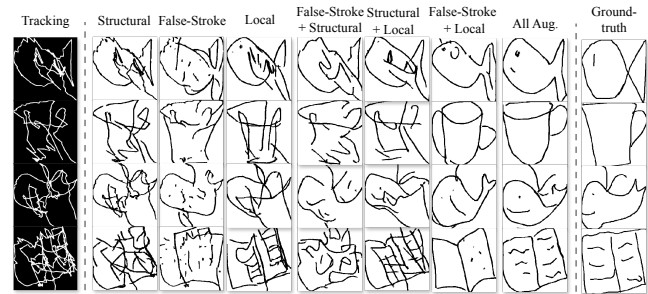

Figure 9: Qualitative results when different combinations of augmentations are applied during training. Each column represents the generated sketches when the model is trained with one combination of augmentations, e.g. Column 2 are the generated sketches when only structural augmentations are applied during training.

In Table 4 we show the similarity scores between the ground-truth and generated sketches with or without prompt on seen or unseen categories. When no augmentation is applied during training, there is little to no performance gain, even with a text prompt. When augmentations are present, in most cases, there is noticeable improvement across all metrics. Such results verify the necessity of our augmentation-based training procedure.

We find that the model tends to rely more on text prompts when working with unseen categories. When prompts are not present, CLIP I2T and CD drop 24.7% and 21.4% respectively on the unseen categories, versus a decrease of 4.9% and 10.3% on the seen categories. In Figure 10, we show

Table 4: Quantitative results on real air drawing dataset with/without prompts. "w/ Aug." refers to with/without augmentations. "w/ P." refers to with/without prompts. "I2I" refers to CLIP Image-Image similarity between tracking/generated and ground-truth sketch, and "I2T" refers to CLIP Image-Text similarity between sketches and texts associated with their class labels.

| w/ Aug. | w/ P. | SSIM (↑) | CD (↓) | LPIPS (↓) | I2I (↑) | I2T (↑) |
|---|---|---|---|---|---|---|
| Seen Categories | | | | | | |
| Tracking | | 0.55 | 32.36 | 0.42 | 0.76 | 0.21 |
| ✗ | ✓ | 0.55 | 31.99 | 0.41 | 0.79 | 0.21 |
| ✗ | ✗ | 0.56 | 32.09 | 0.40 | 0.78 | 0.21 |
| ✓ | ✓ | **0.64** | **25.46** | **0.36** | **0.85** | **0.29** |
| ✓ | ✗ | 0.63 | 26.70 | 0.36 | 0.81 | 0.26 |
| Unseen Categories | | | | | | |
| Tracking | | 0.54 | 33.92 | 0.42 | 0.76 | 0.21 |
| ✗ | ✓ | 0.55 | 33.53 | 0.43 | 0.78 | 0.21 |
| ✗ | ✗ | 0.56 | 33.66 | 0.40 | 0.78 | 0.22 |
| ✓ | ✓ | **0.63** | **24.26** | **0.38** | **0.84** | **0.28** |
| ✓ | ✗ | 0.61 | 30.47 | 0.39 | 0.80 | 0.23 |

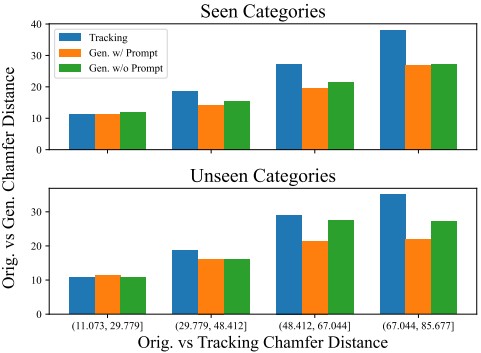

Figure 11: The impact of level of chaos of the tracking on the faithfulness of the generated sketch, based on CD. We divide CD between tracking and ground-truth sketch equally into four bins. Within each bin, we plot the mean of CD between ground-truth sketch, tracking, and generated image with/without prompt.

examples of the model failing to generate the correct sketch when no prompt is present, under unseen categories. Consider the bottom left example of Figure 10: the ground-truth sketch (Ground-truth) represents a snail. With the corresponding text prompt, the model successfully generates a sketch of a snail, following visual cues in the tracking image (Tracking) such as the spiral shape. When no prompt is given, the model falsely, yet reasonably, generates a fan (w/o Prompt) – a category seen during training. In fact, in the absence of a text prompt and given an input from an unseen category, we find that the failure cases tend to be generated sketches of the seen categories.

**Effect of *level of chaos* on conditional input.** In Figure 11, we investigate how the amount of chaos in the input conditioning tracking image, measured by the CD between the ground-truth and tracking images, affects generation. The improvements are more obvious when the CD between the ground-truth and tracking images is large. On the other hand, the CD between ground-truth and generated sketches are almost the same when the tracking image is already close to the ground-truth sketch, indicating that the generation is being faithful to the ground-truth sketch. We also observe that when the level of chaos is high for the unseen categories (bottom right), the performance gain with text guidance is most obvious, suggesting that the model relies more on text guidance for correct generation. Conversely, the model depends less on the text prompts when the input chaos level is low or under seen categories. Such results are also in line with Figure 10 and Table 4 as discussed above.

# 6 Conclusions

In this paper, we tackle the problem of marker-less air drawing by leveraging a spatially-controlled diffusion model. We devise a simple augmentation-based data-free training procedure to learn a mapping from noisy to clean sketches. We collect two hand drawing datasets and verify that the trained model can effectively generate coherent and faithful sketch from an extremely noisy tracking image, and exhibits decent generalizability.

**Limitations and Future Work.** We provide a stepping stone for the proposed task by creating a framework that establishes a correspondence between hand motions and clean, pleasing, and coherent illustrations, and providing datasets to evaluate these tasks. However, this work does not explore the possibility of using hand gestures to create complex, full color images that image diffusers are known for. It also assumes that the desired output sketch is simple and often cartoon-like, as a majority of the Quick, Draw! sketches are drawn with few lines and simple shapes.

**Societal Impact.** Generative models for media are generally prone to misuse which can lead to the spread of inappropriate and harmful content. While this work is based in the sketch domain, there could still be adverse adoption for generating mocking content and bullying, especially by younger users.

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

# A Appendix

## A.1 Details for Sketch Augmentations

**Implementation.**   Below we provide implementation details for each augmentation:

- **Local Augmentation** includes 3 types of sub-augmentations: stroke distortion, random spikes, and jitters. For stroke distortion, we generate a number of wave functions each with random frequency, shift, and amplitude. We aggregate them to form a random curve, whose value is used to distort the ground-truth sketch. Spikes are implemented in two modes sharp spikes and smooth spikes. Sharp spikes are determined by the width and height which are randomly sampled from a normal distribution; smooth spikes are implemented with Bezier curve, and the control points are randomly sampled from a uniform distribution within predefined range along the gradient of the ground-truth sketch. Jitters are simply small perturbations sampled from a pre-defined normal distribution and added to random locations along the ground-truth sketch.

- **Structural Augmentation** also include 3 types of sub-augmentations: sketch-wise distortion & relocation, stroke-wise misplacement, and stroke-wise resize. For sketch-wise distortion& relocation, we randomly shrink and change aspect ratio of the whole sketch and reposition the sketch in the canvas. For stroke-wise misplacement/resize, we randomly move/resize each stroke within a pre-defined range.

- **False Strokes** includes two types: transitional and random false strokes. For transitional false strokes, we simply draw lines between the transition of each stroke; for random false strokes, we randomly draw a number of extra lines on the canvas.

In the above augmentations, misplacement and stroke resize under structural augmentation are mutually-exclusively applied. This is because it is highly probable that the combination of the two will completely destroy the visual cues of the resulting sketch image. For all other augmentations, we set a 50% chance for each to be applied to each sketch sample during training.

## A.2 Training Details

**Held-out categories.**   Because different object categories have different drawing difficulty, sketch and tracking samples belonging to different categories exhibit large variance in statistics (CLIP Image-Text similarity between ground-truth sketch and corresponding text, CLIP Image-Image similarity, Chamfer Distance, or SSIM between ground-truth and tracking image). To allow for fair comparison of the performance between seen and held-out categories, we first partition categories into 10 clusters using K-Means Clustering, and randomly sample one category from each cluster as the held-out categories. The held-out categories are:

$$\{car, face, cow, snail, diamond, candle, angel, cat, grapes, sun\}$$

**Training configurations.**   All training is conducted on two Nvidia H100 GPUs. For finetuning the diffusion model with LORA, we set LORA rank to 4, per device batch size to 16, learning rate to $5e^{-5}$, gradient accumulation steps to 4, and train for 6K steps on the Quick, Draw! dataset. For our augmentation-based ControlNet training, we set per device batch size to 8, learning rate to $2e^{-5}$, gradient accumulation steps to 4, and the proportion of empty prompts to 25%.

## A.3 Comparison on Egocentric Hand Tracking Algorithm

An example of hand landmarking with MediaPipe, OpenPose, and NRSM is shown in Figure 12. We can see that only MediaPipe's hand landmarker comparatively accurate and consistent, while OpenPose often fails to detect landmarks, and NSRM is less accurate. However, even though MediaPipe seems to accurately predict hand landmarks, the resulting tracking image, as shown in Figure 1 and Figure 15, still contains large amount of noise.

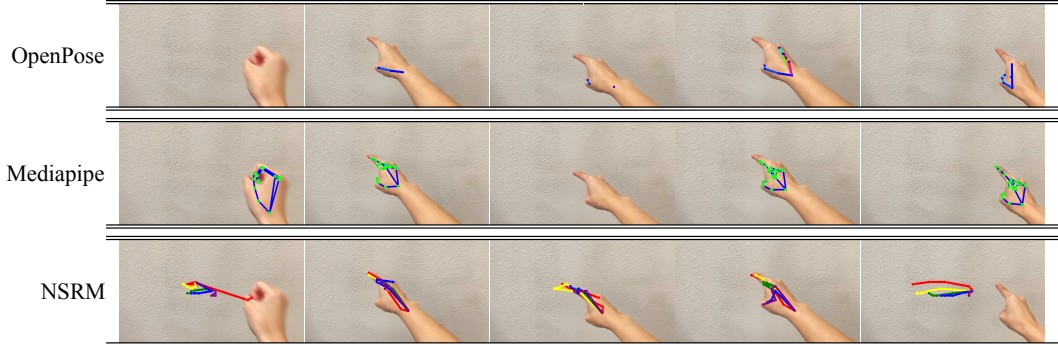

Figure 12: A comparison of hand landmarking done by MediaPipe, OpenPose, and NSRM respectively. MediaPipe in general provides most accurate hand landmarks, while OpenPose often struggling to detect the hand, and NSRM not able to provide accurate landmarks.

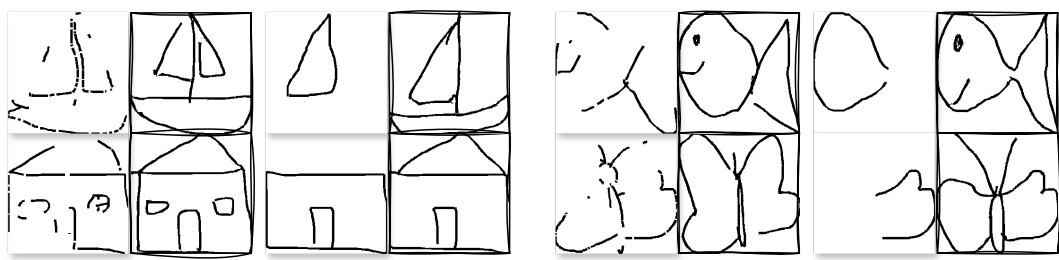

Figure 13: Auto-completion given a partially-drawn sketch in two modes: line segment missing (Column 1&5) and whole strokes missing (Column 3&7). Bordered images are the generated results.

## A.4 Sketch Completion & Text-conditioned styling.

With simple augmentations, we can also perform other sketch-based tasks such as sketch completion and text-conditioned styling. Unlike prior works [39, 38] on sketch completion that require sophisticated architectures or training procedures, we could achieve sketch completion by simply adding random erasure to the set of augmentations during training. In Figure 13, we show two types of sketch completion: completion with small missing line segments (column 1&5), and completion with whole strokes missing (column 3&7). In addition, as shown in Figure 14, we can also utilize the text-conditioning capacity of DMs to prompt for different stroke styles, such as different color and thickness, therefore alleviating the need for fine-grained style control on the user's part.

## A.5 Visual Comparison between Controllable DMs

In Figure 15 we compare the generated sketches between ControlNet and T2IAdapter, when trained under our augmentation-based procedure. We can observe that both ControlNet and T2IAdapter generate visually coherent sketches and generally follow the input tracking image. Nonetheless, by

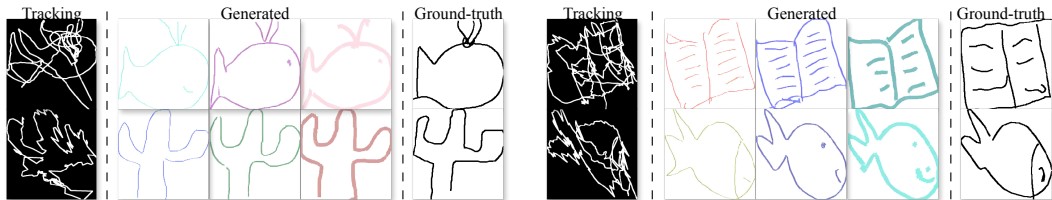

Figure 14: Generation with text instructions for different line colors and brush thickness. Bordered images are the generated results. We use the prompt: "A sketch of a <category>, <color> lines, <thickness> brush."

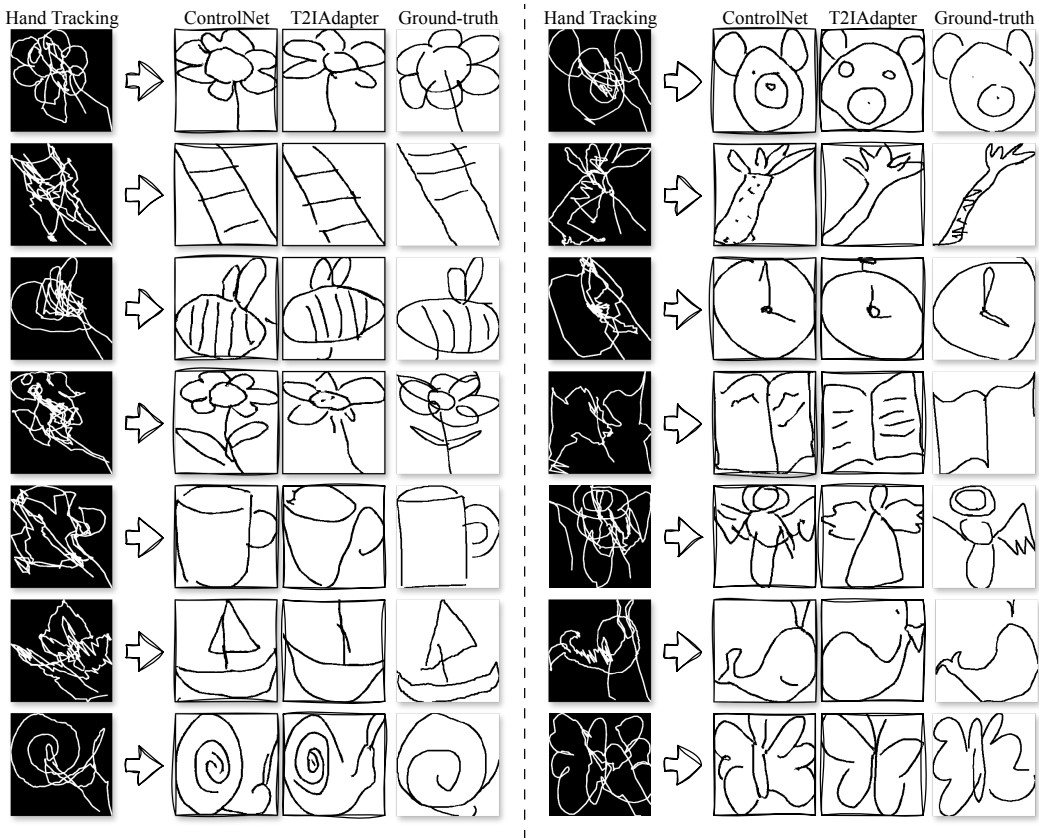

Figure 15: More inference results on ControlNet with our augmentation-based training procedure. We additionally show inference results by using T2IAdapter in our method.

looking into details, ControlNet's generations are more faithful to the visual cues from the tracking images. For example, the butterfly (bottom right) generated by ControlNet resembles the shapes of the wings from tracking image closely, while the generation from T2IAdapter looks much less similar.

