# OpenReview forum: "AirSketch: Generative Motion to Sketch"
_NeurIPS.cc/2024/Conference — NeurIPS 2024 poster_

### Official Review · Reviewer_THqa · 2024-07-07

**Soundness:** 3
**Presentation:** 4
**Contribution:** 2
**Rating:** 6
**Confidence:** 4

**Summary:**

The authors present an interesting application of multiple vision components merged together. They introduce AirSketch to generate sketches directly from hand gestures. They present a self-supervised training procedure and augmentations to with an image diffusion model to generate the realistic sketches. They also present two datasets building on existing datasets and Unity engine. They claim that controllable image diffusion can produce a clear sketch from a noisy input and present a new technique for drawing without markers.

**Strengths:**

The paper is clearly written and covers most of the related works in sufficient depth.
The authors clearly motivate why the problem is important and what could be some clear applications of the idea from an AR/VR perspective (which has become a popular field of research lately).
The authors highlight the lack of hand drawing to sketch datasets and to that end present novel datasets.
The authors also cover the different cases of failures and show that the augmentations can make it robust.

**Weaknesses:**

The idea presented in the paper might be a combination of existing ones and can be easily materialized by using a combination of different models, but it is nevertheless an interesting area.
Very limited description of the hand tracker. The authors should expand on it.
Didn’t find enough motivation for why only Quick, Draw! was used as the starting point?
Is there a reason for choosing Unity over other engines? Can stable diffusion be used to generate the hand gesture videos for the dataset?
The metrics used for evaluations can all be prone to errors. I am not sure if there is sufficient time to get IRB and run human experiments, but that could significantly strengthen the paper.

**Questions:**

Is there anything specific in the model that can solve for jitters? Any particular component of the pipeline that authors think is helping with robustness?
Can something other than Quick, Draw! be used for generating the synthetic dataset?
For ControlNet, the authors use a batchsize of 8. Is that enough to maintain statistics with layers like BatchNorm?
Have the authors looked at more human-like augmentations to help the model better generalize to unseen objects and become more robust to the existing ones? Pls. see Atoms of Recognition, Extreme Image Transformations, etc.

**Limitations:**

The authors find that the model relies on text prompts more when applied to unseen categories. The hypothesis is based on the absolute numbers. The authors should also try and show something like a saliency map for both visual and textual elements of the model to better understand the parts that the model is focusing on.
The authors did not release the code with this submission but promise a future release. Please make sure that the code does accompany the paper for better reproducibility.
The authors also list that there is no need for statistical significance. However, can SSIM not be a good metric to see if the differences between the generated maps and ground truth is significant, when not relying on the human eye?
SSIM and CD by themselves are weak metrics to measure similarity, and are prone to errors. The authors must address that in the text and also tell what might be good ways to mitigate that (i am not asking to devise a new metric, just let the reader clearly know).
The technology can also have a negative societal impact, perhaps not directly but as a derivative of this being applied to another area. The authors should try and address those areas.

---

> ### Author Rebuttal · Authors · 2024-08-06
>
> #### **Limited description of the hand tracker**
> We thank the reviewer for pointing this out! While we did discuss hand trackers in the appendix (A.3), we will add a more explicit description to the main body of the paper.
>
> #### **Why only Quick, Draw! was used as the starting point?**
> 1. When drawing in the air without controllers or headsets, quick and convenient everyday sketches are likely to resemble the simplistic sketches found in the Quick, Draw! dataset, rather than the more elaborate sketches in the TUBerlin dataset.
> 2. All prior works on sketch generation, such as [a,b,c], only consider the Quick, Draw! dataset.
> 3. The Quick, Draw! dataset is the biggest sketch dataset, and is the only dataset with timestamp information for each point in the drawing. We needed this information to create the synthetic dataset, where the hand moves at a speed accurate to how the sketch was truly drawn.
>
> #### **Is there a reason for choosing Unity over other engines? ​​Can stable diffusion be used to generate the hand gesture videos for the dataset?**
> Generating hand gesture videos with Stable Diffusion would be much harder than simply using a graphics engine, especially since it is prone to incorrectly generating hands (even despite recent improvements). Unity is a very mature graphics engine and does the job for us, but we are totally open to suggestions from the reviewer for future work.
>
> #### **IRB and human experiments**
> Unfortunately, we won't have enough time within the rebuttal period to get that done. Our institution has an extensive IRB protocol.
>
> #### **Is there anything specific in the model that can solve for jitters?**
> We account for jitters (as a result of the hand tracking and human tremors) by introducing the jitter augmentation during training. The model then learns to clean up jitters when they are encountered in the input.
>
> #### **Any particular component of the pipeline that authors think is helping with robustness?**
> The part of the pipeline that we think helps the most with the robustness are our suite of augmentations, which helps our model learn what aberrations in the sketches need to be removed. We made sure to carefully develop our augmentations to account for common errors in sketching that are encountered in our use-case, whether it be from human error, the hand tracking, or simply the nature of the task.
>
> #### ** other datasets than Quick, Draw! for generating the synthetic dataset?**
> Unfortunately since Quick, Draw! Is the only dataset providing timestamp associated with drawing coordinates, it is the only available dataset for creating synthetic drawing videos.
>
> #### **For ControlNet, the authors use a batch size of 8. Is that enough to maintain statistics with layers like BatchNorm?**
> ControlNet’s standard training regime has a batch size of 4, so our batch size of 8 is likely not too small.
>
> #### **Have the authors looked at more human-like augmentations to help the model better generalize to unseen objects and become more robust to the existing ones?**
> We thank the reviewer for the suggestions to include more “human-like” augmentations and bringing up interesting topics in Atoms of Recognition and Extreme Image Transformations[c,d]; we will consider them for improving robustness in our future work. Meanwhile, the way we conduct our augmentations are designed to simulate a comprehensive range of “human-like” errors found in the noisy sketches produced by our air drawing.
>
> #### **Statistical significance**
> While it is too computationally intensive to conduct training multiple times, we re-conduct our main result in Table [1] from the paper by performing inference 10 times on each sample with different seedings, and report the mean&std shown in the table below.
> |w/ Aug.|SSIM (↑)|CD (↓)|LPIPS (↓)|CLIP I2I (↑)|CLIP I2T (↑)|
> |-|-|-|-|-|-|
> ||||**Seen Categories**|
> |✘| 0.55±0.03$e^{-3}$ | 31.92±0.12$e^{-3}$|0.41±4.0$e^{-3}$|0.77±0.78$e^{-3}$|0.21±0.41$e^{-3}$|
> |✓| **0.64**±0.14$e^{-3}$ |**25.13**±0.22|**0.36**±1.0$e^{-3}$|**0.84**±1.2$e^{-3}$ | **0.29**±0.93$e^{-3}$|
> ||||**Unseen Categories**|
> |✘|0.55±0.08$e^{-3}$| 33.53±0.31$e^{-3}$|0.41±3.8$e^{-3}$|0.78±0.84$e^{-3}$|0.21±0.51$e^{-3}$|
> |✓|**0.63**±0.45$e^{-3}$|**24.26**±0.71|**0.38**±1.9$e^{-3}$|**0.85**±2.8$e^{-3}$|**0.28**±1.7$e^{-3}$|
>
> We also conducted a Paired T-test for significance. However as the means have a clear margin and std is low, the p-values are all low in the scale of 1e-14. Since this is not a rigorous test, we are not including the results here.
>
> #### **The authors should also try and show something like a saliency map for both visual and textual elements of the model to better understand the parts that the model is focusing on.**
> Figure [4] (right) in the rebuttal PDF shows the visualization of ControlNet’s hidden state throughout the reverse diffusion steps. We can clearly observe that while the model trained with augmentation (bottom) is able to infer the clean sketch contour (t>=16), the one trained without augmentation (top) fails to do so.
>
> #### **Data & Code release**
> We would like to assure the reviewer that we will absolutely release the code and datasets with the paper for better reproducibility, once the paper is accepted to a venue.
>
> #### **Negative societal impact**
> We thank the reviewer for bringing this up. The only potential negative impact we could think of is the use of our model for generating malicious content. However since we focus on sketch generation, the impact is much less than a regular image generation model. However, we would love to hear what the reviewer thinks, so we can include it in the final version of our paper.
>
> [a] A Neural Representation of Sketch Drawings
>
> [b] Sketch-pix2seq: a model to generate sketches of multiple categories
>
> [c] Controllable stroke-based sketch synthesis from a self-organized latent space
>
> [d] Atoms of recognition in human and computer vision
>
> [e] Extreme Image Transformations Improve Latent Representations in Machines

---

> > ### Comment · Reviewer_THqa · 2024-08-11
> >
> > I would like to thank the authors for their time to answer the questions. I would like to see some clarifications on the following:
> >
> > IRB: Given human subjects, the authors should try to obtain IRB, even if it is extensive (for a reason).
> >
> > Jitters: Can the authors provide some visualizations/insights into how the model is doing that?
> >
> > Human-like augmentations: Can the authors clarify why are the Atoms of Recognition and Extreme Image Transformations not useful here for robustness or generalization?
> >
> > Negative societal impact: There could be an adverse adoption of this work for generating mockery/meme videos, specially in the teen category. What do the authors think?
> >
> > I look forward to seeing the thoughts and/or clarifications on the above points in the manuscript.
> >
> > Thanks.

---

> ### Author Response · Authors · 2024-08-11
> **Reply to Reviewer THqa**
>
> Thank you again for your time and feedbacks!
>
> #### **IRB: Given human subjects, the authors should try to obtain IRB, even if it is extensive (for a reason).**
>
> We thank the reviewer for mentioning. We have initiated IRB and will include in our final version of the paper.
>
> #### **Jitters: Can the authors provide some visualizations/insights into how the model is doing that?**
>
> During training, we apply augmentations (jitters, false strokes, distortions etc.) to the sketch and ask the model to reconstruct the original sketch, thereby forcing the model to learn to remove noise and distortions. In Figure 6 (main paper) we show ablations on models trained with different augmentations. As can be seen from the 1st, 2nd and 4th image where jitters exist on the top edges of these images, models trained with structural and false-stroke augmentations are not able to remove the jitter, whereas the model trained with Local (e.g. jitter) augmentation successfully removes the jitter.
>
> In addition, we run additional inferences on jittered sketches. In the resulting generations, all the jitters are removed as expected. We have sent the anonymous sharing link to AC per policy, who shall in turn release it.
>
> #### **Human-like augmentations: Can the authors clarify why are the Atoms of Recognition and Extreme Image Transformations not useful here for robustness or generalization?**
>
> Due to the length limit we were not able to expand on this topic in the above reply. Below is a more detailed discussion:
>
> Atoms of Recognition [a] discovers the phenomenon where humans fail sharply to recognize objects with a tiny degradation from Minimal Recognizable Configurations(MIRC), which is not observed on computational models. Instead of improving robustness/generalizability of computational models, this work mainly focuses on the pitfall of human recognition instead of machine’s, which seems to be tangential to our focus. However, it could be interesting to conduct sketch-based MIRC for machines and see if any improvement can be derived, which we will leave for future work.
>
> Extreme Image Transformations (EIT) [b] utilizes global transformations (e.g. grid/full shuffle, color flatten etc.) to improve classification/detection robustness and generalizability by forcing the model to learn global representation. In our case, given black and white sketches, only grid shuffle is applicable, and we indeed considered grid shuffle (Jigsaw) as one of the augmentation initially, but the resulting sketch often became incomprehensible even to human eyes – because of the high versatility and the absence of color and textual information in sketches.
>
> In terms of “Human-like augmentations”, the augmentations we implemented are intended to simulate “human-like” errors made during air-drawing, while both Atoms of Recognition and EITs are not “Human-like” augmentations.
>
> #### **Negative societal impact: There could be an adverse adoption of this work for generating mockery/meme videos, specially in the teen category. What do the authors think?**
>
> We thank the reviewer for raising the point. In general generative AI can result in negative impact if misused, and we agree in our case the negative impact can be specifically to teenagers. We will add it to the final version of the paper.
>
> [a] Atoms of recognition in human and computer vision
>
> [b] Extreme image transformations affect humans and machines differently

---

### Official Review · Reviewer_Gs1n · 2024-07-08

**Soundness:** 3
**Presentation:** 3
**Contribution:** 2
**Rating:** 5
**Confidence:** 4

**Summary:**

The paper presents a technique for generating raster handwritten sketches based on the tracking of the hand (from egocentric video), with the target application scenario of sketching in AR/VR.

The training is done mostly based on Quick, Draw! dataset of sketches combined with generated hand videos using Unity (5k samples). In addition, 500 real videos were collected by the authors for the purposes of training and evaluation.

The model is ControlNet-based (most experiments using Stable Diffusion XL).

The authors show qualitatively that this technique allows for generating sketches that are quite close to the ground truth, and can be extended to sketch completion. They also show the importance of providing a text prompt describing the object category that needs to be generated. The also show the importance of using the data augmentation when generating the synthetic dataset, to ensure that synthetic sketch data is close to the real data obtained by hand tracking algorithm.

**Strengths:**

1. Originality. The paper presents a first-of-its-kind approach and a first-of-its-kind dataset to train the model on.
2. Clarity. The writing is mostly clear and easy to follow.
3. Quality. The ablation study is well done and highlights the importance of the main decisions made by the authors.

**Weaknesses:**

The main weakness of the approach is the quality of the analysis, which brings under question the significance of the technique.

In particular, the authors use Stable Diffusion model LoRA-tuned on Quick, Draw dataset (lines 209-210) for all their experiments. While the evaluation is done on a held-out set of Quick, Draw classes (appendix, line 521), it seems that the tuning could allow leakage of the set of classes to the model generation - suggesting that instead of the ability to generalize to unseen classes, the model actually simply memorized  the class set.

Another similar issue is the technique for the selection of the held-out set of classes (appendix, lines 514-520) where the held-out classes were selected from the K-Means clustering of the classes in Quick, Draw - meaning that they are intentionally similar to the ones used during training.

More generally, the unanswered question of the applicability of this technique to anything beyond the given set of 50 common sketching classes is the main weakness of the paper - generalization to truly unseen sketch classes, ability of SDXL and CLIP to deal with more complex objects or multiple objects on the same image is unclear.

The second weakness is the limited reproducibility of the approach - the authors have not made even the synthetic dataset public (even though they claim to intend to), citing (appendix, line 659) "As of the submission of the paper, the new datasets presented in this paper need to be cleaned to ensure personal information of the creator isn’t released unintentionally" - which is unlikely to be an issue for synthetic data.

The third weakness is the set of evaluation metrics selected by the authors. While it does a good job of highlighting the importance of decisions made by the authors, for me as the reader, the most important missing piece of information about the quality of this approach that I am interested in is "how often does the output of the model actually depict the object of the same class as the user intended to draw?" - which could be obtained by running a high-enough quality classifier on the generated data or looking to the closest neighbour in CLIP embedding space with all of the class categories.

**Questions:**

1. Could you please clarify / show whether SDXL that has only been tuned on Quick, Draw classes that are not used for evaluation, could perform equally well?

2. How does the method perform on other sketches that are further away from the training data (ex not the cluster centers for clustering of the training data)?

3. Could you please clarify the exact dataset used for evaluation? Was it only comprised of real videos, or of synthetic training samples as well? What was the train-test split?

I am willing to increase my score if the authors could provide answers to Q 1/2 and planning to decrease it if there is insufficient evidence that the model can perform well beyond the training Quick, Draw classes.

**Limitations:**

The authors have generally addresses the limitations of their work (with the exception of the data release, see above).

---

> ### Author Rebuttal · Authors · 2024-08-06
>
> #### **Weakness 1.1: Finetuning samples leaking**
> *-- “While the evaluation is done on a held-out set of Quick, Draw classes (appendix, line 521), it seems that the tuning could allow leakage of the set of classes to the model generation - suggesting that instead of the ability to generalize to unseen classes, the model actually simply memorized the class set.”*
>
> In Figure[3] we run inference on less-common classes that are not seen during both augmentation-based training and LoRA finetuning, and we can clearly observe that the model maintains its performance over these unseen classes.
>
> #### **Weakness 1.2: Regarding K-Means clustering for partitioning seen/unseen classes**
> *-- “... the held-out classes were selected from the K-Means clustering of the classes in Quick, Draw - meaning that they are intentionally similar to the ones used during training.”*
>
> We would like to clarify:
> * The K-Means clustering is based solely on evaluation data (our collected synthetic/real hand videos), not training data.
> * We perform the clustering over the order of chaos of classes – how different is the trajectory image from the ground-truth (SSIM/CD), and how similar is it to its corresponding text label (CLIP I2T). We are *not* using it to intentionally split semantically similar classes into seen/unseen classes (e.g. assigning “dog” to seen and “gray dog” to unseen).
> * We are *not* sampling unseen classes from one particular cluster that “resembles the training data”, but uniformly from all clusters, such that the averaged baseline statistics between seen/unseen are similar and can thus be compared.
>
> Different classes from the Quick, Draw! dataset have vastly different qualities. Without such a procedure it will be impossible to quantitatively draw comparisons two sets of classes. In addition, as we show in our ablations, order of chaos has a clear impact on model performance. By drawing unseen classes uniformly from cluster centers, we can perform such ablations (e.g. Figure [8] main paper) in different order of chaos.
>
> We apologize for any ambiguous wording that causes the confusion, and will improve the expression in the final draft of the paper.
>
> #### **Weakness 1.3: Applying to anything out of the 50 classes/more complex objects?**
> We show that our model is indeed capable of generalizing out of the seen classes and to generate more complex sketches. For results on classes out of the 50 classes please see Figure[3] in rebuttal PDF and response under Weakness 1.1. For results on more complex objects please see Figure [1] in rebuttal PDF for results on TUBerlin dataset and general response 1.
>
> #### **Weakness 2: Synthetic dataset has not been made public.**
> Our datasets will become available when the paper is accepted to a venue! It is a very common practice for researchers to wait to release code and datasets until their paper is accepted, to avoid having their work misappropriated by other researchers prior to publication; we hope the reviewer understands. We will absolutely release all code and datasets once the paper has been accepted into a venue.
>
> #### **Weakness 3: Evaluation metrics**
> *-- “The third weakness is the set of evaluation metrics selected by the authors…I am interested in is "how often does the output of the model actually depict the object of the same class as the user intended to draw?"*
>
> In the paper we indeed report the CLIP image-to-text similarity (CLIP I2T) to measure how likely is the output depicting the intended class. However if the reviewer believes a different metric is needed, we are happy to provide further evaluation.
>
> #### **Q1: Could you please clarify / show whether SDXL that has only been tuned on Quick, Draw classes that are not used for evaluation, could perform equally well?**
> Yes, please see Figure[3] in the rebuttal PDF and response under Weakness 1.1.
>
> #### **Q2: How does the method perform on other sketches that are further away from the training data (ex not the cluster centers for clustering of the training data)?**
> We hope that the clarification about the K-Means selection and the evaluation on the unseen classes/datasets answers this question.
>
> #### **Q3: Could you please clarify the exact dataset used for evaluation? Was it only comprised of real videos, or of synthetic training samples as well? What was the train-test split?**
>
> We used the collected synthetic and real hand motion datasets only for evaluation. In the main paper (e.g. Table[1]) we report the testing accuracy separately under synthetic and real dataset. The training samples were all augmented-clean sketch pairs from the Quick, Draw! dataset and do not overlap with the evaluation samples.

---

> ### Author Response · Authors · 2024-08-13
> **Does our reply address your concern?**
>
> Dear reviewer,
>
> Thank you again for your feedbacks! As the rebuttal period is approaching the end today, could you please check if our reply addresses your concerns?
>
> Thank you for your time and feedbacks,
>
> Authors

---

### Official Review · Reviewer_gzU1 · 2024-07-10

**Soundness:** 3
**Presentation:** 3
**Contribution:** 3
**Rating:** 6
**Confidence:** 3

**Summary:**

This paper addresses a new task: sketch generation from marker-less air drawing. The authors trained a spatially conditioned diffusion model to generate sketches from noisy hand tracking results and text prompts. During training, they devised an augmentation-based training procedure.

**Strengths:**

1. Good paper writing. Easy to follow.
2. This paper addressed a new task: marker-less sketch generation, which is useful in AR/VR applications.
3. Experiments are performed on both real and synthetic datasets.
4. The idea of "soft" spatial constraint is novel, which harnesses the capability of image generation model to de-noise the output of hand tracking algorithms.

**Weaknesses:**

1. The model may overfit on the sketches from the Quick, Draw! dataset, as the diffusion model is finetuned on this dataset.
2. The ability to generate simple geometries without certain semantics is not explored. For example, can the model generate a simple curve following the user's drawing?

**Questions:**

1. Could the authors please provide more results that is not from the Quick, Draw! dataset?
2. Could the authors please generation results of simple geometry drawing?

**Limitations:**

1. The input of this model is a complete drawing of the user, which means the user cannot see their half-way drawing. This is not favorable in many applications.
2. This method only takes the hand tracking algorithm's output as input. It ignores the rich information from the video, which may help to achieve more precise control over drawing.

---

> ### Author Rebuttal · Authors · 2024-08-06
>
> #### **Model overfitting on Quick, Draw! Dataset**
> In Figure[3] in the rebuttal PDF we show inference results on unseen classes during both augmentation-based training and LoRA finetuning, where our method still maintains its performance.
>
> On the other hand, we agree with the reviewer that due to the finetuning, the model is “overfit” to the dataset’s style. However, fine-tuning diffusion models to a specific style for consistent outputs is a common practice in the image generation community and is not generally regarded as a limitation.
>
> #### **Results from a different dataset**
> In Figure [1] in the rebuttal PDF, we show results from additional experiments on the TUBerlin dataset, which contains more complicated sketches compared to Quick, Draw! Dataset. As can be seen in Figure [1], our method is able to generate faithful and coherent sketches given more complex objects.
>
> #### **Simple geometries**
> In Figure[2] in the rebuttal PDF we show examples from simple geometries shapes such as triangles, circles and squares, and the model is able to draw them faithfully.
>
> #### **Can’t see halfway drawing**
> *-- "The input of this model is a complete drawing of the user, which means the user cannot see their half-way drawing. This is not favorable in many applications."*
>
> First, we have shown that our model is capable of auto-completing sketches with partial input (Figure[10] Appendix), and therefore the input is not limited to complete drawing. From a research perspective, interactive sketching and editing in a generative manner is indeed interesting, which we will consider in our followup works. From a practical AR/VR standpoint, certain software engineering designs can also be incorporated to enable users seeing half-way drawing/performing interactive sketching. While we do not focus on the engineering side in this work, we will be happy to engage the reviewer about this during the discussion period.
>
> #### **Lose wealth of information from the video**
> *-- “This method only takes the hand tracking algorithm's output as input. It ignores the rich information from the video, which may help to achieve more precise control over drawing.”*
>
> We fully agree with the reviewer’s observation, which in fact aligns with our initial approach: translating video content directly to sketch under an Encoder-Decoder framework. However we found it very hard to learn meaningful trajectory information from video encoders such as VideoMAE. Additionally, there is no large-scale, high quality dataset that includes the rich information found in videos and yet has highly detailed sketches as targets. Creating such a dataset is also time consuming. However, we do agree that by considering additional information such as video features or motion velocities, the generation quality can be improved, which we leave for future work.

---

> > ### Comment · Reviewer_gzU1 · 2024-08-12
> >
> > Thank the authors for their response. I think all my concerns are addressed. I hope that the authors will include all the additional results and discussion in their final manuscript. For now, I will keep my initial score.

---

> > > ### Author Response · Authors · 2024-08-12
> > > **Thanks Reviewer gzU1**
> > >
> > > Thank you for taking time to go through our paper and rebuttal. We will certainly incorporate all the additional results and discussion in the final version.

---

### Official Review · Reviewer_V9Hw · 2024-07-10

**Soundness:** 3
**Presentation:** 3
**Contribution:** 2
**Rating:** 4
**Confidence:** 5

**Summary:**

The authors investigate the problem of sketch generation from the finger's motion trajectory, which is interesting. To make it, the authors adopt a standard ControlNet pipeline, while highlighting the importance of data augmentation (adding noise to clean sketches to mimic the hand-tracking image). Hence the major contribution of this paper is the tweak that makes ControlNet applicable for generating clean sketches from the continuous messy finger trajectories.

**Strengths:**

- The authors repurpose ControlNet to produce a clean sketch from a severely deformed and line-messy input, which is useful for finger motion to sketch generation and could be potentially valuable for other related HCI applications.
- The results look promising, and the application of sketch-completion and text-instructed stroke styling are interesting.
- The experimental results are extensive and validate the effectiveness of the proposed augmentation strategies.

**Weaknesses:**

- The major concern is about the novelty of the technical contribution. Personally, the proposed augmentation methods are more like practical tricks for better cleaning up messy sketches.
- The methods of accepting raster-format input for sketch generation should be also compared by training these models with the same augmented sketches, such as [a][b][c].
- The practical usage seems limited since: (1) the proposed method is only validated on simple sketches, so no way to justify whether any scalability issue when facing complex scenarios; (2) the user probably needs to edit (e.g., deleting lines) while drawing, how to deal with it?
- The collected real air-drawing dataset is very small.

[a] Chen, Yajing, et al. "Sketch-pix2seq: a model to generate sketches of multiple categories."

[b] Yang, Lan, et al. "Sketchaa: Abstract representation for abstract sketches."

[c] Zang, Sicong, Shikui Tu, and Lei Xu. "Controllable stroke-based sketch synthesis from a self-organized latent space."

**Questions:**

Please refer to the weaknesses.

**Limitations:**

The major limitation may be the limited practical usage as I listed in weaknesses.

---

> ### Author Rebuttal · Authors · 2024-08-06
>
> #### **Concern about the novelty of the paper**
> Thank you for your feedback. While augmentation is a well-known method, it has never been applied for sketch generation, and our finding that it can be used as an effective self-supervised pre-training task for airsketch is in our opinion a non-trivial contribution. Further, we would like to reiterate that the contribution of our work is not only about an implementation ‘trick’, but also
> 1. a novel task of generative motion to sketch, with a dataset. There is no other work, including those cited by the reviewer, that achieves the task nor was the task previously proposed. We believe it will open the doors to many interesting applications by bringing generative modeling into the AR space.
> 2. A concrete baseline method that has been thoroughly evaluated and analyzed, with performance that we believe will be useful for future research and applications.
> 3. A novel way of viewing and exploiting controllable diffusion models.
> We are happy to discuss this further.
> #### **Comparison to raster-format input for sketch generation**
> We thank the reviewer for mentioning these works. As per requested, we run Sketch-Pix2Seq on a subset of 10 classes. For each class, we train a separate Sketch-Pix2Seq model by first training 50K steps without augmentation, followed by another 50K steps with augmentation. Results are summarized in Table [1] below and figure [4](left) in the rebuttal PDF. In particular, we find that while Sketch-Pix2Seq produces semantically coherent sketches, it often cannot identify and follow visual cues from the messy tracking image, as shown in Figure [1](left). It thus leads to a noticeable improvement in terms of CLIP I2I/I2T, but a marginal improvement over SSIM/CD.
>
> |                   | SSIM (↑) | CD (↓)       | LPIPS (↓)     | CLIP I2I (↑) | CLIP I2T (↑) |
> |-------------------|----------|--------------|---------------|--------------|--------------|
> | Tracking          | 0.5      | 32.36        | 0.42          | 0.76         | 0.21         |
> | Sketch-Pix2Seq    | 0.58±1.7$e^{-3}$ | 30.19±0.60 | 0.38±1.5$e^{-3}$  | 0.82±0.78$e^{-3}$ | 0.26±0.41$e^{-3}$ |
> | Ours              | **0.63**±0.21$e^{-3}$ | **25.45**±0.26 | **0.36**±1.1$e^{-3}$ | **0.84**±2.0$e^{-3}$ | **0.29**±0.86$e^{-3}$ |
>
> In addition, we also want to point out that works like [a][c] could only fit a model on one or few classes at a time and struggle to generalize beyond – while both [a][c] aim at learning across multiple classes, their maximum number of categories is only 5, which is far less than what is needed in our scenario. Therefore we did not include them for benchmarking.
>
> #### **Practical usage**
> (1) *-- "the proposed method is only validated on simple sketches, so no way to justify whether any scalability issue when facing complex scenarios"*
> * We run additional experiments on the TUBerlin dataset. Results are summarized in Figure [1] in the rebuttal PDF, which shows our method is able to generate faithful and coherent sketches given more complex objects.
> * All previous works in sketch generation [a][c] also only consider simple sketches from Quick, Draw! dataset.
>
> (2) *-- "the user probably needs to edit (e.g., deleting lines) while drawing, how to deal with it?"*
>
> While sketch editing is indeed an interesting topic, we believe this may be out of scope of this work and could become an interesting future direction.
>
> As our work is the first to study motion to sketch in a generative manner, we acknowledge (Line 41-45, Introduction) there is a large space of exploration and improvement. Meanwhile, to deploy such models into production-ready AR/VR systems, certain software engineering designs are still needed, which is yet not our primary concern in this work.
>
> #### **Size of the dataset**
> We acknowledge that the real dataset is small for training tasks. However, these are difficult datasets to collect, especially for a first work in this task. We are still in the process of expanding the dataset to support future work. On the other hand, this underlines the necessity of our augmentation-based training, which is self-supervised and does not require labeled data.
>
>
> [a] Chen, Yajing, et al. "Sketch-pix2seq: a model to generate sketches of multiple categories."
>
> [b] Yang, Lan, et al. "Sketchaa: Abstract representation for abstract sketches."
>
> [c] Zang, Sicong, Shikui Tu, and Lei Xu. "Controllable stroke-based sketch synthesis from a self-organized latent space."

---

> > ### Comment · Area_Chair_o1X7 · 2024-08-12
> > **Could you check if the rebuttal addresses your concern or not, please?**
> >
> > Dear Reviewer V9Hw, thank you for your time in reviewing this paper. It would be great if you could check if the rebuttal addresses your concern or not. Thank you in advance.

---

> ### Comment · Reviewer_V9Hw · 2024-08-13
>
> I appreciate the authors for the detailed responses. Most of my concerns have been addressed.
>
> However, I still think it is weak on the technical side, and this work is more like a cute application paper.
>
> That said, I like the angle of this work and I believe it would attract sufficient attention to encourage further research works in this direction. So I am open and will be happy to follow other reviewers' lead if they consistently feel it should be accepted.

---

> > ### Author Response · Authors · 2024-08-13
> >
> > Thank you for acknowledging our response and the perspective of our work. Concerning the technical aspects, we would like to emphasize the following points:
> >
> > 1. We believe that model complexity should not be the primary criterion for evaluating the merit of a study. Many existing methods are effective despite their simplicity. Our method: (a) avoids additional complexity, allowing for better reproducibility and easier hyper-parameter tuning (b) is self-supervised, eliminating the need for labeled data, (c) produces high-quality results that are robust to significantly distorted trajectory images and generalizable to unseen classes, and (d) is evaluated through extensive experiments. Given the novelty of our proposed task, we believe a simple yet effective approach shall serve as a solid baseline, upon which more complex designs can be built as future works.
> >
> > 2. The contribution of this work extends beyond the technical implementation. It also introduces a novel task that treats motion-to-sketch from a generative perspective, along with newly collected datasets. As the reviewer noted, this contribution is likely to attract considerable attention and stimulate further research in this area.

---

### Author Rebuttal · Authors · 2024-08-06

We thank all reviewers for spending time reading our paper and providing insightful feedback. We appreciate reviewers finding our proposed task and approach novel, interesting, and useful (V9Hw, gzU1, Gs1n), and the experiments being extensive and convincing (V9Hw, Gs1n, THqa). We address each of the reviews separately below but provide some common responses here.

### Generalizability on unseen classes
Figure[3] in the rebuttal PDF shows inference results on less-common classes not seen during both LoRA fine-tuning and augmentation-based training. We could clearly observe that the model maintains its performance over these unseen classes.

### Experiments on TUBerlin dataset
We run additional experiments on the TUBerlin dataset following the same training procedure from the main paper, and results are shown in Figure[1] in the rebuttal PDF. Our method produces faithful and coherent results with much more complicated sketches compared to Quick, Draw!, demonstrating its scalability to more complex scenarios (Reviewer V9Hw, gzU1).

---

### Decision · Program_Chairs · 2024-09-25

**Decision:**

Accept (poster)

**Comment:**

The main weakness of this paper is centralized in the technical contribution of this paper, while the author emphasizes the task of generative motion to sketch as the main contribution of this paper. Overall, the reviewers appreciate the newly created task in this paper. Given these concerns raised by reviewers, the author provided detailed and professional rebuttals.
I hope the author can revise the paper according to the suggestions from reviewers to make their paper stronger for the camera-ready.